# Post-Diagnostic Diet Quality and Mortality in Females with Self-Reported History of Breast or Gynecological Cancers: Results from the Third National Health and Nutrition Examination Survey (NHANES III)

**DOI:** 10.3390/nu11112558

**Published:** 2019-10-23

**Authors:** Nena Karavasiloglou, Giulia Pestoni, David Faeh, Sabine Rohrmann

**Affiliations:** 1Division of Chronic Disease Epidemiology, Institute for Epidemiology, Biostatistics and Prevention (EBPI), University of Zurich, Hirschengraben 84, CH-8001 Zurich, Switzerland; 2Cancer Registry Zurich and Zug, University Hospital Zurich, Zurich, Switzerland, Vogelsangstrasse 10, CH-8001 Zurich, Switzerland; 3Health Department—Nutrition and Dietetics, Bern University of Applied Sciences, Falkenplatz 24, CH-3012 Bern, Switzerland

**Keywords:** dietary patterns, cancer survivors, cohort, adults, mortality

## Abstract

High quality diets are associated with favorable disease and mortality outcomes in various populations; little and conflicting information is available for female cancer survivors. We investigated the association of post-diagnostic diet quality with mortality in female cancer survivors. Data from 230 women with a previous breast, or gynecological (i.e., ovarian, cervical or uterine) cancer diagnosis in the Third National Health and Nutrition Examination Survey were analyzed. The Healthy Eating Index (HEI) and the Mediterranean Diet Score (MDS) were calculated based on a 24-hour dietary recall interview. Cox proportional hazards regression models were used to calculate multivariable-adjusted hazard ratios (HR) and 95% confidence intervals (CI). Higher HEI score was associated with lower mortality (HR_HEI_ total = 0.97, 95% CI: 0.95–0.98, 1 unit increase), but the association for MDS failed to reach statistical significance (HR_MDS_ total = 0.87, 95% CI: 0.74–1.04). In subgroup analyses, a statistically significant inverse association was observed between the HEI and mortality; for the MDS, no statistically significant association was apparent. Higher post-diagnostic HEI score was inversely associated with mortality in female cancer survivors, suggesting a protective effect when adhering to the diet captured by the HEI. Additional studies are required in order to investigate underlying mechanisms of the mortality-adherence association.

## 1. Introduction

Even though breast cancer and gynecological cancers are the most common cancers in women worldwide [1], little is known about the post-diagnostic habits women adopt and if they are associated with mortality. A recent report indicated that while no recommendations could be made for cancer survivors, there are indications that healthy body weight, physical activity, and dietary factors post-diagnosis may be associated with better survival [2]. This shortage of evidence on the influence of post-diagnostic habits is particularly important since the number of cancer survivors is growing and evidence-based recommendations to prolong survival are needed.

Few studies have reported on the association between post-diagnostic lifestyle and survival in cancer patients and their results are contradicting. Decreased mortality risk in breast cancer survivors who consumed at least five daily portions of fruit and vegetables and were physically active was reported [3]. Older female cancer survivors with higher adherence scores to the 2007 World Cancer Research Fund/American Institute for Cancer Research guidelines on body weight, physical activity, and diet had lower mortality of all causes; lower all-cause mortality was also reported for women who only followed the dietary recommendations [4]. On the other hand, high vegetable, fruit, and dietary fiber intake and low fat consumption was not associated with reduced mortality risk in early stage breast cancer survivors in a different study [5].

The Healthy Eating Index (HEI) is associated with lower risk of chronic diseases and mortality in different populations [6,7]. Higher diet quality (assessed by the HEI) was associated with lower all-cause, cardiovascular disease, and cancer mortality risk in postmenopausal women [8] and a meta-analysis reported similar findings [9]. Additionally, higher post-diagnostic diet quality was associated with lower mortality risk, especially mortality from non-breast cancer causes, in breast cancer survivors [10].

The Mediterranean diet is associated with lower mortality in older populations [11,12,13,14], as well as in coronary heart disease [15], cardiovascular disease [16] and diabetic patients [17]. A recent meta-analysis reported no association between adherence to the Mediterranean diet and cancer-specific mortality in cancer patients including data from four studies [18]. Furthermore, no association between the Mediterranean diet with either all-cause mortality or non-breast-cancer-related mortality was reported in breast cancer survivors. However, a statistically significant inverse association with death from non-breast cancer causes was shown in women who reported low physical activity [19].

Based on the possible association of a higher quality post-diagnostic diet with decreased mortality in cancer survivors and the limited number of studies that have reported on it, we believe that the association of post-diagnostic diet quality with mortality in female cancer survivors warrants further investigation.

## 2. Materials and Methods

### 2.1. Population

In this study, data from the Third National Health and Nutrition Examination Survey (NHANES III) were used, due to the long follow-up of the participants. The NHANES III was a nationwide survey, conducted between 1988 and 1994 in the United States. Participants were interviewed and underwent physical examination in a mobile examination center. Detailed description of the methodology of the NHANES III is described elsewhere [20]. NHANES III data are publically available and can be accessed online [21]. Therefore institutional review board approval and oversight were not required for our study.

In this study, only female participants who had self-reported a previous breast, ovarian, cervical or uterine cancer diagnosis were included in the analyses. Participants were defined as cancer survivors if they answered “yes” to the question, “Has a doctor ever told you that you had other cancer?” In following questions participants had to specify the site of their cancer diagnosis and their age at cancer diagnosis (“How old were you when you were first told you had other cancer?”). Participants who also self-reported a skin cancer diagnosis (*n* = 26; answered “yes” to the question, “Has a doctor ever told you that you had skin cancer?”), with missing information on either of the exposure variables (*n* = 4) or any of the confounding variables (*n* = 50) were excluded from the analyses (Figure 1). The final study population included 230 participants. Of these, 110 were breast cancer survivors (BCa) and 120 were gynecological cancer (i.e., ovarian (*n* = 19), cervical (*n* = 54), and uterine cancer (*n* = 47); GynCa) survivors.

### 2.2. Outcome Ascertainment

The outcome of interest was all-cause mortality. Probabilistic linkage of the NHANES III with the National Death Index, maintained by the National Center for Health Statistics, was used to obtain mortality information. Detailed description of the linkage was previously described [22]. Previous studies have reported accurate ascertainment of participants’ death by the National Death Index [23,24,25]. Follow-up time was defined as the time from completion of the NHANES III questionnaire (in months) until death from any cause or end of follow-up (December 31, 2011), whichever came first.

### 2.3. Assessment

Dietary information was obtained via a 24-hour dietary recall interview. Data collection was scheduled as such as to include interviews all days of the week and throughout the year. During an interview by trained interviewers, participants were requested to report their food and beverage consumption of the past 24 hours. Additional information including (but not limited to) the quantities, recipes, and place of eating was also recorded. The nutrient values were calculated using recipe ingredient nutrient values found in the US Department of Agriculture’s (USDA) Survey Nutrient Database Nutrient Files. The dietary information of the 24-hour dietary recall interview was used to calculate the diet quality indices.

To assess moderate to vigorous physical activity NHANES III participants were asked how frequently (but not for how long) they performed leisure time exercise or physical activities in the past month. Their answers were then transformed to “times per week” using the conversion factor 4.3 weeks per month.

### 2.4. Healthy Eating Index

The HEI, developed by the USDA, includes 10 equally weighted distinct components and the sum of each participant’s score in the components constitutes their individual score. Components 1–5 estimate a participant’s conformity to the recommendations by the Food Guide Pyramid for the grain, vegetable, fruit, milk, and meat groups. Components 6–9 capture total fat, saturated fat, total cholesterol, and sodium dietary intake. Finally, component 10 estimates the variety in a participant’s diet. The score can range from 0 to 100 and the higher a participant’s score on the HEI, the better the diet according to the Dietary Guidelines for Americans and the Food Guide Pyramid [20]. Participants were dichotomized based on their HEI (poor (HEI score < 70); good (HEI score ≥ 70)).

### 2.5. Mediterranean Diet Score

The MDS [13] assesses the adherence to the Mediterranean diet by including the consumption of 9 components (legumes, vegetables, fruit and nuts, cereals, fish and seafood, meat and meat products, dairy products, the ratio of monounsaturated to saturated fats and alcohol). The components of the MDS were composed by linking the 24-hour dietary recall data with My Pyramid Equivalents Database for USDA survey food codes. The daily consumption of each of the components (in servings/100 g/day) was calculated for every participant. The median consumption for each component was estimated in our study population and then participants were scored with 0 or 1 for each component based on the relation of their consumption to the study median. Participants with consumption of legumes, vegetables, fruit and nuts, cereals, and fish and seafood above the median received 1 point (per component), whereas consumption of meat and meat products, dairy products and a ratio of monounsaturated to saturated fats lower than the median received 1 point (per component). Finally, participants with alcohol consumption of 5–25 g per day (approximately 0.3 to 2 drinks) received 1 point. The sum of all the points was calculated as each individual’s MDS. The MDS can range from 0 to 9. Participants were divided into two groups based on their score; participants with MDS 0–4 were considered to be “non-adherers”, whereas participants with MDS 5–9 were considered to be “adherers”.

### 2.6. Statistical Analysis

Baseline categorical data were expressed as percentages and continuous data as means and standard errors of the mean (SEM). The association between each diet quality index and mortality was assessed using Cox proportional hazards regression models. Regarding confounder adjustment, the first model (Model 1) was adjusted only for age at completion of the NHANES III questionnaire and race/ethnicity (Non-Hispanic white, Non-Hispanic black, Mexican-American, Other). The second model (Model 2) was additionally adjusted for several a priori determined confounders based on the existing literature including: time between cancer diagnosis and completion of the NHANES III questionnaire (continuous, years), socioeconomic status (SES; categories based on the poverty income ratio (PIR), poor/near poor (PIR < 2), middle income (2 ≤ PIR < 4), higher income (PIR ≥ 4), or unknown; similar to Suresh et al. [26]), marital status (married/living together, never married/widowed, divorced/separated, or unknown), body mass index (BMI; continuous, kg/m^2^), physical activity (continuous, times of moderate to vigorous physical activity per week), smoking status (never, former, or current smoker), self-reported prevalent chronic diseases at baseline (type 2 diabetes, heart attack, congestive heart failure or stroke; yes, no, or unknown), daily energy intake (continuous, kcal/day) and history of menopausal hormone therapy use (never or no information, ever user). Model 2 was further adjusted for alcohol intake (frequency of beer, wine, and hard liquor intake; times per month) in the analyses for the HEI but not for the MDS, since alcohol intake is already included within this index. The results were presented as hazard ratios (HR) and 95% confidence intervals (CI).

Baseline characteristics were compared between female breast or gynecological cancers survivors and female participants of the NHANES III without a self-reported cancer diagnosis. Sensitivity analysis was performed excluding alcohol consumption from the MDS.

Statistical analyses were performed using SAS (version 9.3, SAS Institute Inc., Cary, NC, USA) software and significance levels were set at a = 0.05. Sampling weights, adapted according to our total study population and sub-groups, were used in all analyses to account for the complex survey design and survey non-response.

## 3. Results

Overall mean time between cancer diagnosis and completion of the NHANES III questionnaire was 10.4 years (SEM 0.7; breast cancer survivors: 8.6 years, SEM: 0.7; gynecological cancer survivors: 12.0 years, SEM: 1.0). Breast cancer survivors completed the NHANES III questionnaire at an older age and were less likely to be current smokers compared to gynecological cancer survivors (Table 1). Breast cancer survivors were more likely to report higher income and had slightly higher BMI compared to gynecological cancer survivors. Female breast or gynecological cancer survivors included in this project were less physically active and more likely to be current smokers compared to female NHANES III participants who did not report history of cancer (Table 2). Regarding their dietary habits, female breast or gynecological cancer survivors had slightly higher HEI score and increased fruit and vegetable consumption compared to female participants who did not report history of cancer.

The association between the diet quality indices and mortality, after an overall mean follow-up time of approximately 16 years, is shown in Table 3. The number of deaths recorded until the end of follow-up was 121. In continuous models, the HEI was statistically significantly associated with lower mortality (HR_HEI total_ = 0.97, 95% CI: 0.95–0.98, HR_HEI BCa_ = 0.97, 95% CI: 0.95–0.99, HR_HEI GynCa_ = 0.92, 95% CI: 0.89–0.96, in the fully adjusted models, per 1 unit increase in the HEI score). In categorical models, high diet quality (corresponding to HEI ≥ 70) was statistically significantly associated with mortality risk in the total study population (HR_HEI good_ vs. _poor total_ = 0.43, 95% CI: 0.29–0.64). Sub-group analyses revealed similar results for gynecological cancer survivors (HR_HEI good_ vs. _poor GyCa_ = 0.20, 95% CI: 0.10–0.43) and breast cancer survivors (HR_HEI good_ vs. _poor BCa_ = 0.49, 95% CI: 0.25–0.97).

The MDS was not associated with mortality risk in the total study population (HR_MDS total_ = 0.87, 95% CI: 0.74–1.04, in the fully adjusted model, per 1 unit increase in the MDS). Subgroup analyses revealed similar not statistically significant associations in both groups (HR_MDS BCa_ = 0.97, 95% CI: 0.82–1.16; HR_MDS GynCa_ = 0.77, 95% CI: 0.57–1.04). In categorical models, the categories of the MDS were not statistically significantly associated with mortality risk in the total population (HR_MDS Adherers vs. Non-Adherers_ = 0.67, 95% CI: 0.41–1.11) or the subgroups. Excluding alcohol consumption from the MDS did not materially modify our results (data not shown).

## 4. Discussion

In our study of female cancer survivors, post-diagnostic diet quality, as assessed by the HEI, was associated with reduced mortality. In subgroup analyses, the statistically significant inverse association was evident in both breast and gynecological cancer survivors. For the MDS, a non-statistically significant inverse association was observed overall and in the subgroups.

Even though statistically significant associations were observed between the HEI and mortality, the association between the MDS and mortality was not statistically significant. A possible explanation for the lack of statistically significant associations is that very few participants in our study had high MDS. The MDS was based on the diet observed in European populations living around the Mediterranean region. On the other hand, the HEI was specifically designed to capture the diet quality of the US population (e.g., the first five components reflect conformity to the Food Pyramid Guide for the US population). This could suggest that the dietary pattern adopted by the US population differs from the Mediterranean diet and that the MDS may not be able to accurately capture the diet of this population. Given the multiple health benefits associated with the Mediterranean diet, several interventions were proposed in order to facilitate the adoption of the Mediterranean diet by the US population [27].

An inverse association between higher-quality diets (captured either by the HEI or the MDS) and mortality risk in disease-free populations was suggested previously [6,8,11,28,29]. Additionally, several studies have reported better prognosis in cancer survivors [30] and/or specifically female cancer survivors who adhered to a higher quality diet after diagnosis [4,10,31,32]. However, there are also studies suggesting that post-diagnostic diet and/or post-diagnostic diet quality are not associated with survival [5,19,33]. Our findings on the HEI are in line with the former studies and provide additional evidence in favor of a higher quality post-diagnostic diet as a modifiable risk factor.

A mechanism potentially explaining our findings could be mediated through inflammation. Breast cancer survivors who adopted higher quality diets post-diagnosis exhibited lower C-reactive protein (CRP) levels, compared to those with lower quality diets [34]. A positive association between CRP levels and mortality in disease-free individuals [35], as well as in cancer patients [36] is reported in the literature. Furthermore, improved survival after invasive breast cancer diagnosis was reported in women consuming diets with low inflammatory potential [37]. Diet with high inflammatory potential was previously associated with higher all-cause mortality risk in participants of the NHANES III [38]. Given that the diet corresponding to higher HEI score is considered a diet with low inflammatory potential, the inverse association we observed could be mediated through lower CRP concentrations. Another possible mechanism explaining our findings could be through meeting the micronutrient and mineral dietary requirements. Lower quality diets may be insufficient/deficient in various protective micronutrients and minerals and/or be abundant in potentially harmful agents. Long-term low micronutrient and mineral intake could be associated with worse health outcomes, including mortality [39,40].

The lifestyle differences and the lack of diet quality differences between female NHANES III participants without a self-reported cancer diagnosis and the female breast or gynecological cancer survivors included in this study (Table 2) could indicate that other lifestyle modifications, e.g., increasing physical activity or quitting smoking, may be easier to implement and to sustain over time compared to dietary modifications. Thus, further guidance and support may be necessary for female cancer survivors to adhere to a healthier, higher-quality diet post-diagnosis.

Our study had several strengths. The prospective study design and the long follow-up time allowed us to establish a clear period between disease onset, post-diagnostic dietary habits, and mortality. Additionally, the detailed information collection at baseline allowed us to adjust for various lifestyle factors that may influence mortality.

However, this study also has several limitations. The small sample size of female breast or gynecological cancer survivors may not have allowed the detection of a significant association between the MDS, and mortality. Additionally, our results were based on only one assessment of the dietary habits and may not reflect the true or long-term habits of individual participants. However, both diet quality indices were validated in previous studies and seem to adequately capture peoples’ dietary habits. As in any observational study, we cannot exclude the possibility of residual confounding. We cannot exclude the possibility that the female cancer survivors that participated in the NHANES III and in our study could have been healthier than female cancer survivors that did not participate in the NHANES III. Due to the small number of gynecological cancer survivors we were not able to perform analyses separately for every cancer type. It is plausible that some cancer cases (e.g., ovarian) were detected early since otherwise, patients’ survival is very low. Finally, since information regarding disease severity or treatment was not available, we were not able to account for them in our analyses. Yet, given the long time between cancer diagnosis and mortality, cancer treatment and severity are unlikely to severely affect our results. Future projects should aim to capture this information and include them in the analyses as covariates.

In conclusion, higher post-diagnostic diet quality, as assessed by the HEI, was inversely associated with mortality in female cancer survivors, suggesting a protective effect when adhering to the diet captured by the HEI. Additional studies are required in order to verify our findings and to investigate underlying mechanisms of the mortality-adherence association.

## Figures and Tables

**Figure 1 nutrients-11-02558-f001:**
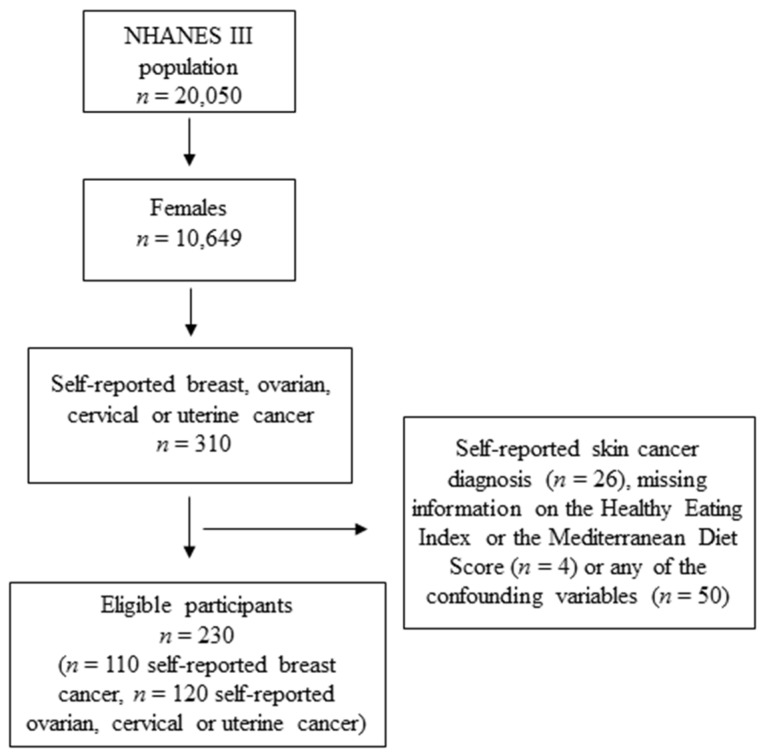
Flowchart of the study participants.

**Table 1 nutrients-11-02558-t001:** Socioeconomic characteristics and participants’ behaviors at completion of the NHANES III questionnaire according to their cancer diagnosis and combined ^1^.

	Total Study Population (*n* = 230)	Breast Cancer Survivors (*n* = 110)	Gynecological Cancer Survivors (*n* = 120)
Age at completion of the NHANES III questionnaire (years, mean, SEM)	54.4, 1.5	62.4, 1.6	47.4, 1.6
Age at diagnosis (years, mean, SEM)	44.0, 1.3	53.7, 1.5	35.4, 1.1
Time from the completion of the NHANES III questionnaire until the end of the follow-up (months, mean, SEM)	192.2, 6.9	170.4, 9.6	211.6, 5.7
Race/Ethnicity, %			
Non-Hispanic white	89.8	91.6	88.2
Non-Hispanic black	6.8	5.0	8.4
Mexican-American	2.0	1.5	2.4
Other	1.4	1.9	0.9
Marital status, %			
Married/living together	61.8	60.0	63.3
Never married/widowed	21.0	27.0	15.7
Divorced/separated	17.2	13.0	21.0
Socioeconomic status, %			
Poor or near poor	31.0	25.3	36.0
Middle income	36.6	38.9	34.6
Higher income	25.7	26.9	24.6
Unknown	6.8	8.9	4.8
Smoking status, %			
Never	38.6	42.5	35.1
Former	33.7	40.5	27.7
Current	27.7	16.9	37.3
Moderate and vigorous physical activity (times/week), %			
0	20.8	22.5	19.3
0 to <2	25.8	29.2	22.9
2 to <4	22.4	13.9	29.9
4 to <6	6.5	8.1	5.1
>6	24.4	26.2	22.8
Body mass index (kg/m^2^, mean, SEM)	25.8, 0.5	26.4, 0.5	25.3, 0.5
History of menopausal hormone therapy use, %			
Never or No information	55.7	54.5	56.8
Ever	44.3	45.5	43.2
Prevalent chronic diseases at baseline, %			
Yes	14.3	14.7	14.0
No	83.5	81.1	85.6
Unknown	2.2	4.2	0.4
Healthy Eating Index, %			
Poor (score <70)	60.0	49.7	69.0
Good (score ≥70)	40.0	50.3	31.0
Mediterranean Diet Score, %			
Non-Adherers (score 0–4)	75.7	73.7	77.4
Adherers (score 5–9)	24.3	26.3	22.6

^1^ SEM: Standard Error of the Mean.

**Table 2 nutrients-11-02558-t002:** Lifestyle characteristics of females without a self-reported cancer diagnosis vs. females with self-reported breast or gynecological cancers in the NHANES III ^1^.

	Females without Self-Reported Cancer (*n* = 8424)	Females with Self-Reported Breast or Gynecological Cancers (*n* = 230)
Age at completion of the NHANES III questionnaire (years, mean, SEM)	42.1, 0.5	54.4, 1.5
Race/Ethnicity, %		
Non-Hispanic white	73.8	89.8
Non-Hispanic black	12.6	6.8
Mexican-American	5.2	2.0
Other	8.4	1.4
Marital status, %		
Married/living together	59.2	61.8
Never married/widowed	27.8	21.0
Divorced/separated	12.9	17.2
Unknown	0.1	0.0
Socioeconomic status, %		
Poor	14.3	11.2
Near poor	20.7	19.8
Middle income	35.0	36.6
Higher income	23.2	25.7
Unknown	6.9	6.8
Smoking status, %		
Never	56.4	38.6
Former	18.0	33.7
Current	25.6	27.7
Moderate and vigorous physical activity (times/week), %		
0	18.1	20.8
0 to <2	24.9	25.8
2 to <4	16.9	22.4
4 to <6	7.4	6.5
>6	32.7	24.4
Body mass index (kg/m^2^, mean, SEM)	26.3, 0.2	25.8, 0.5
History of menopausal hormone therapy use, %		
Never	20.8	43.5
Former	7.2	29.3
Current	7.3	15.0
Unknown/No information	64.6	12.2
Breastfeeding, %		
Yes	38.6	44.4
Healthy Eating Index (mean, SEM)	64.2, 0.3	66.3, 1.0
Mediterranean Diet Score (mean, SEM)	3.5, 0.0	3.4, 0.1
Total energy intake (kcal/day, mean, SEM)	1805.2, 15.1	1,741.4, 57.1
Fruit (times/month, mean, SEM)	28.5, 0.6	32.9, 3.2
Vegetables (times/month, mean, SEM)	66.5, 1.0	72.8, 5.5
Red meat (times/month, mean, SEM)	19.7, 0.4	19.8, 1.5

^1^ SEM: Standard Error of the Mean.

**Table 3 nutrients-11-02558-t003:** The association between diet quality indices and mortality for all participants and according to their cancer diagnosis ^1^.

	Total Study Population (*n* = 230)		Breast Cancer Survivors (*n* = 110)		Gynecological Cancer Survivors (*n* = 120)	
	Model 1	Model 2	Model 1	Model 2	Model 1	Model 2
Healthy Eating Index	0.97 (0.95–0.99)	0.97 (0.95–0.98)	0.97 (0.95–0.99)	0.97 (0.95–0.99)	0.95 (0.93–0.97)	0.92 (0.89–0.96)
Poor (<70)	1.00	1.00	1.00	1.00	1.00	1.00
Good (≥70)	0.45 (0.30–0.67)	0.43 (0.29–0.64)	0.49 (0.33–0.72)	0.49 (0.25–0.97)	0.29 (0.16–0.51)	0.20 (0.10–0.43)
Mediterranean Diet Score	0.84 (0.73–0.97)	0.87 (0.74–1.04)	0.89 (0.77–1.03)	0.97 (0.82–1.16)	0.73 (0.56–0.94)	0.77 (0.57–1.04)
Non-adherers (0–4)	1.00	1.00	1.00	1.00	1.00	1.00
Adherers (5–9)	0.52 (0.33–0.82)	0.67 (0.41–1.11)	0.47 (0.29–0.76)	0.78 (0.47–1.32)	0.63 (0.33–1.21)	0.49 (0.18–1.37)

^1^ Healthy Eating Index: Model 1: Adjusted for age at completion of the NHANES III questionnaire (continuous, years) and race/ethnicity (non-Hispanic white, non-Hispanic black, Mexican-American, other); Model 2: Additionally, adjusted for time between diagnosis and completion of the NHANES III questionnaire (continuous, years), body mass index (continuous, kg/m^2^), marital status (married/living together, never married/widowed, divorced/separated, unknown), socioeconomic status (poor/near poor, middle income, higher income, unknown), history of menopausal hormone therapy use (never user or no information, ever user), smoking status (never, former, or current smoker), self-reported prevalent chronic diseases at baseline (type 2 diabetes, heart attack, congestive heart failure or stroke; yes, no, or unknown), alcohol consumption (continuous, times per month), daily energy consumption (continuous, kcal/d) and moderate to vigorous physical activity (continuous, times of moderate to vigorous physical activity per week). Mediterranean Diet Score: Model 1: Adjusted for age at completion of the NHANES III questionnaire (continuous, years) and race/ethnicity (non-Hispanic white, non-Hispanic black, Mexican-American, other); Model 2: Additionally, adjusted for time between diagnosis and completion of the NHANES III questionnaire (continuous, years), body mass index (continuous, kg/m^2^), marital status (married/living together, never married/widowed, divorced/separated, unknown), socioeconomic status (poor/near poor, middle income, higher income, unknown), history of menopausal hormone therapy use (never user or no information, ever user), smoking status (never, former, or current smoker), self-reported prevalent chronic diseases at baseline (type 2 diabetes, heart attack, congestive heart failure or stroke; yes, no, or unknown), daily energy consumption (continuous, kcal/d) and moderate to vigorous physical activity (continuous, times of moderate to vigorous physical activity per week).

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
