# Peer review of "Post-Diagnostic Diet Quality and Mortality in Females with Self-Reported History of Breast or Gynecological Cancers: Results from the Third National Health and Nutrition Examination Survey (NHANES III)"

_nutrients, 2019, doi:10.3390/nu11112558_

Round 1
Reviewer 1 Report
This study assessed the association between 'post-diagnosis diet quality' and mortality in females with a history of 4 major female cancers using NHANES III data. I have two main concerns about this study - one for exposure measure (post-diagnosis diet quality) and another for potential significant confounders.
First, the data RE diet quality were obtained when the participants joined the study. Thus, we are not sure whether the diet patterns changed after diagnosis for the study population (these participants with a history of the 4 selected cancers). If the diet patterns did not change since the cancer diagnosis among them, then you are not measuring the effect of 'post-diagnosis' diet quality on all cause mortality. It is also important to know that changing a person's diet is very difficult as diet is very personal, involving deeply entrenched behavior related to family, culture, community and even values.
Second, as the outcome measure is all cause mortality, many chronic health conditions at baseline should be considered in the comparison (if they were available), such as diabetes or cardiovascular diseases. If such information is not available during the very long follow-up (average 16 years), many other factors could bias the association found in this study.
Some specific comments:
I have concern about the term used 'female cancer survivors', which could be misunderstood as survivors from any cancer (except for non-melanoma skin cancer). Why only select these specific cancers? The sample size could increase substantially if they included females with a history of other cancer, such as colorectal cancer, etc. socioeconomic status measure needs more explanation and a reference to support its use. Table 1 should have included the comparison of the differences between those with a self-report cancer diagnosis and those without, rather than the comparison of the two cancer groups. Also in Table S1, the title of the 2nd column may not be accurate. You are comparing two groups - one with report of these cancers and another without. In the flowchart, I cannot understand why 26 skin cancer patients were included in the first place and then later were excluded. The significance of a result should be determined by a statistical test, and p-values from such a test should be provided in Table 2. In the second sentence of Discussion, the author concluded that the association was not statistical significant for breast cancer in subgroup analysis based on categorical model (compared good with poor diet quality). I don't believe this is correct, because different cut-off point for good or poor may reach different conclusion. I also think that the results of treating diet quality as a continuous measure are more accurate and should be used as main results.
Author Response
"Please see the attachment."

Reviewer 2 Report
Comments: Thank you for asking me to review this manuscript submission. This is an interesting analysis on an important subject that suggests that post-diagnostic dietary quality is associated mortality in women this a history of breast and gynecologic cancers. An important recommendation is that “further guidance and support may be necessary for female cancer survivors to adhere to healthier, higher quality diets post diagnosis.
Background:
Should include brief description of 5-year mortality rates for types of cancers described in this paper Use of general statement of “post-diagnostic diet quality” is vague — could refer to time-frame immediately after diagnosis, 5 years after diagnosis or follow-up after even longer period. Survivor behavior is likely different at each stage. Should you address this possibility in the introduction? CRP and inflammation are addressed for the first time in the discussion. Suggest introducing this possible mechanism/hypothesis in the background otherwise comment in discussion comes out of the blueMethods:
Should state why you used NHANES III data —presumably to establish the longitudinal component of the study —this type of statement will help the reader appreciate your study design and choice of NHANES cycles. Line 73—may want to end paragraph with statement “therefore IRB approval and oversight were not required." Use of terms “participants”, “recruitment", etc., are not really appropriate for this study design; subjects participated in NHANES, they were not recruited for and they did not participate in this study, however, their data was used for this analysis Would like additional description of how NHANES participants were matched to mortality information — given that NHANES data is de-identified, how did you do this; what does Probabilistic linkage of NHANES III to National Death Index records really mean? I think a brief description would help the uninformed reader. Line 86, add “participant” after NHANES III Line 87, do you mean National Center for Health Statistics? — is “statistics” missing? Line 89, add “NHANES” before questionnaire; “completion of the NHANES questionnaire” Suggest adding statement of potential maximum duration from NHANES III survey completion to end of follow-up, e.g., 18-24 years (if I calculated this correctly) Line 92-93—statement of “use of automated data collection instrument” is somewhat confusing. Was the automated data collection instrument a script that was electronically prompted to the interviewer? Otherwise could be confused with the ASA24 system that has been used to collect dietary data for NHANES in later cycles Line 95—confirm and if true, state that a single 24 dietary intake was used to generate data for HEI and MDS otherwise could be confused with two dietary recalls that were used in later cycles. Line 120, may want to provide an example of the number of grams in a typical glass of wine or other alcoholic beverage to provide some perspective for the reader.(e.g., there are x grams of alcohol in a typical 6 oz glass of red wine) Line 130-131- statement that time between cancer diagnosis and inclusion in the study — may want to include that this question was included in the survey in the earlier methods statement. Also, instead of referring to “inclusion in the study” you should probably state, “completion of the NHANES survey”.Results:
Table 1 title, Line 182: Use of the terms/phrases “recruitment” and “inclusion in the study” are misleading. Participants weren’t recruited into this study. Suggest changing this term in the text of the Table 1 to “at time of NHANES survey completion”. Likewise change “Age at study entry” in first row of table Data on physical activity was provided in Table 1 but not discussed in methods. Would like to have seen Table with summary data of dietary intake — individual components of HEI and HEI scores and components of MDS and total score--very general information is in the supplementary table but given the focus on intake for this study additional summary information is warranted. Would like to have seen summary data of overall follow-up time from completion of NHANES survey completion included in the table. Figure S1: Would like to see "Eligible participants" further subdivided into Breast, Ovarian, Uterine, etc. classifications with sample size designatedDiscussion:
Line 199-200; reference to “few participants in our study had high MDS” needs to be supported by summary data in results section. Lines 204-206: This statement needs to be supported by summary data CPR or anti-inflammatory diet should be introduced in background section; this discussion comes out of the blue Lines 234-236: this statement is important and needs to be emphasized Line 238: modify “allowed us for adjusting for various…” to “allowed us to adjust for various…” Line 243: modify to “may not reflect the true or long-term habits…” add “true” Lines 217, 246 247: change “that” to “who” when referring to a person or people. This change should be made throughout Lines 244-245: Important to know that NHANES dietary analysis is appropriate to summarize dietary intake for population-based studies but it is not intended to assess individual dietary quality Line 257: Recommend changing “its corresponding diet” to “The dietary components of the HEI...Author Response
"Please see the attachment."

Round 2
Reviewer 1 Report
The authors addressed my concerns well and I have no further comments. Thus, I recommend the paper to be accepted.
Author Response
Thank you very much.